# Prophylaxis and Remediation for Future Pandemic Pathogens—(Lessons from a Post-COVID World)

**DOI:** 10.3390/microorganisms10122407

**Published:** 2022-12-05

**Authors:** Mark E. Obrenovich, Moncef B. Tayahi, Caryn L. Heidt, Steven N. Emancipator

**Affiliations:** 1Department of Veteran’s Affairs Medical Center, Research Service, Cleveland, OH 44106, USA; 2Department of Chemistry, Case Western Reserve University, Cleveland, OH 44106, USA; 3The Gilgamesh Foundation for Medical Science and Research, Cleveland, OH 44116, USA; 4Department of Medicinal and Biological Chemistry, College of Pharmacy and Pharmaceutical Sciences, University of Toledo, Toledo, OH 43614, USA; 5Department of Chemistry, Cleveland State University, Cleveland, OH 44115, USA; 6Department of Biological and Environmental Sciences, Cleveland State University, Cleveland, OH 44115, USA; 7Department of Computer Science and Electrical Engineering, University of Cincinnati, Cincinnati, OH 43145, USA; 8Department of Chemical Engineering, Michigan Technological University, Houghton, MI 49931, USA; 9Health Research Institute, Michigan Technological University, Houghton, MI 49931, USA; 10Department of Pathology, Case Western Reserve University, Cleveland, OH 44106, USA

**Keywords:** SARS-CoV-2, personal protective equipment, PPE prophylactic, COVID-19, pandemic, isoelectric point, virucidal coatings, COVID preparedness

## Abstract

Since influenza and coronaviruses are currently deadly and emerging threats worldwide, better treatment, remediation and prevention options are needed. In that regard, a basic understanding of severe acute respiratory syndrome (SARS)-CoV-2/COVID-19 (*Betacoronaviridae*) and other viral pathogen mechanisms of transmission are expected. Unfortunately, unprecedented, and growing distrust of vaccines and even masks or personal protective equipment (PPE) in the United States and elsewhere presents itself as an added challenge. We postulate that development of improved and highly effective prophylactic measures, together with new life-saving therapies that do inhibit or otherwise treat infection of SARS-CoV-2, influenza and other viral pathogens, could be an adjunct measure to globally protect vulnerable individuals from pandemic threats. In this review, we share what we learned from the past COVID experience to offer a multifactorial and improved approach to current and future pandemic infections or threats using low-cost means.

## 1. Introduction

Engineering approaches, including those for improved personal protective equipment (PPE) and remediation of fomite transfer has been one focus for dealing with the recent SARS-CoV-2 virus pandemic and its community spread. Once it emerged in 2019 in Wuhan City, Hubei Province, China, the COVID pandemic has become endemic in most of the world. Early in the pandemic, SARS-CoV-2 rapidly spread throughout the world and, as pandemics trend to do, has infected all continents in just a matter of months. The pandemic has wrought financial havoc and taken a significant, but not unprecedented human toll. Yet, this virus demonstrates extreme resilience; it was highly mutated and transmissible. One might say the virulence of this pandemic was overall mild and the world was lucky that it was not more virulent. However, luck is a poor strategy going forward and, in this paper, we lay out our approach to past and future respiratory pandemic prophylaxis.

The specter of another pandemic looms large, since we know there are numerous new variants to older strains and new emerging pathogens. A recent strain of H1N1, swine flu (G4 EA H1N1), has recently emerged in China and may continue to spread throughout China or elsewhere. This new virus strain and other SARS variants have all the essential hallmarks of another candidate pandemic virus [1]. Fortunately, we now have the new mRNA vaccine technology. When this became available, it was transformative and effective. These vaccines are among the best defense we can expect. A hierarchy of protection approaches adapted from CDC guidelines are shown in Figure 1. Wed discussed all the approaches but focused the project report on personal approach [2].

Unfortunately, these approaches are not fully accepted, and can be costly or take months to years to develop. Therefore, the world population requires more rapid and universal protective approaches, and that next logical step would be to design, and wear improved or enhanced PPE (EPPE). Moreover, employing highly effective virus-trapping and eradicating measures including: (i) engineering better air handling and filtration, (ii) improving PPE and (iii) general virucidal approaches are the first line and least expensive means to managing future pandemics world-wide. In this review, we propose concepts and share the concepts that we formulated to remediate the threat at that very precarious time in our collective histories. We are sharing the technology while retaining all rights and intellectual property this disclosure should entails in the effort to share what we learned.

To be clear, elimination of fomites, which involves basic physical removal of pathogens, is accomplished through surface disinfectant application or handwashing coupled with engineering controls, like UV-C germicidal light and air filtration [3]. The focus of this manuscript is to utilize select environmental engineering approaches to target airborne transmission, over fomite transmission, which is one part of an overall strategy to limit infection risk indoors and on surfaces. When combined with instruction for correct PPE removal or handwashing technique, which are termed administrative or educational controls and are increasingly less effective as we examine the individual transmission level and personal infection control. Prophylactic controls are unfortunately the first line of defense, but they are the least effective approaches to preventing infection. This is where our discussion focusses in offering novel ways to improve PPE, which include masks, gowns and respirators (see Figure 1).

## 2. Materials and Methods

### 2.1. Applied IEP Findings to Mask Design

We have devised an additional approach to test mask design, its coating effectiveness or to predict either adhesion or repulsion. In that regard, an accurate Iso-electric point (IEP) is key to understanding that process to address a more global and functionally simple approach to personal protection equipment (PPE). For IEP measurements, we designed the chemical force microscopy (CFM) method to experimentally measure the IEP in relevant physiological conditions and as a single particle (27). This method reduces the necessity for highly purified samples, which is required for many IEP measurements.

### 2.2. Testing Chamber

To accomplish testing, we first devised a material testing chamber (see Figure 2), where our materials and coatings for PPE could be directly tested. We could use actual attenuated virus or the surrogate large, enveloped virus MS2 phage, which is often used in research settings to mimic viral exposure and transmission [4]. This surrogate is more resistant than non-enveloped viruses like SARS. Eradicating viruses and their variants like, influenza A and B or MERS, SARS and other coronavirus would have to interfere at one of the following nodes: (1) at the level of adhesion and receptor binding by targeting virion attachment to the cell surface, (2) endocytosis by preventing internalization of the virus into the cell, (3) preventing or fostering decapsidation and release of viral ribonucleoprotein, (4) block cytoplasmic transport and nuclear import, (5) preventing RNA transcription and replication, (6) inhibiting nuclear export and protein synthesis, (7) preventing packaging and assembly of progeny, and (8) preventing lytic mechanisms or budding and release from the cell membrane. Targeting the first step in viral infection negates the tendency to focus on downstream processes. Further, the first step is the simplest to accomplish and would be the quickest and least expensive approach. The aim of the testing chamber is to test all aspects of viral load binding repulsion inhibition or other aspects of viral interaction using first a surrogate virus, and then placed in a biosafety level (BSL) 3 situation to test the actual pathogen.

### 2.3. Electronic Wearable Device Design

Next, we devised a complete wearable electronic devices or headgear setup for the repulsion of viral particles using electronic conductive mesh made of parallel wires properly spaced to effectively conduct current unidirectionally. The power sources were are supplied from a mobile phone or a compact power bank via a standard mini universal serial bus to provide constant power for a tunable electric field with various strength at given input applied voltages.

### 2.4. Virus and Surrogate Viral Modeling

SARsCoV2 Virus isolates Working with non-BSL methods and the testing chamber, we obtained heat-inactivated and gamma-irradiated isolates from BEI Resources (Manassas, VA, USA) for SARS-CoV-2 (USA-WA1/2020).

## 3. Background and Significance

### 3.1. Physical Transmission Engineering Controls

The routes of transmission for SARSCoV2 (coronaviridiae) and other respiratory diseases, largely involve inhalation of small airborne “respiratory droplets”, which can be large (over 100 microns) or small (less than 100 microns) in size. The result has been deadly, and the consequences of the pandemic continue to this day and is now an endemic corona virus. Infection occurs through direct contact with infected people and to a lesser extent by contact with contaminated surfaces. COVID-19 is a respiratory virus and the significance of viral transmission via small airborne microdroplets, also known as aerosols, has been extensively reviewed. We briefly discuss transmission in the context of SARS coronavirus-2 pandemic as background to the presenting problem [5]. Uncertainties do remain regarding the relative contributions from different transmission pathways, but it is without question that the best first line of personal defense is though more effective PPE. Existing evidence is sufficiently strong to warrant greatly improved prevention and more effective germicidal components added to different forms of PPE. Moreover, supply chain issues and ongoing global shortages of personal protective equipment remain a problem [6]. When speaking of viral loads, we are stating a size range from 1–4μm to in size. In indoor, air SarsCov2 viral load contains a range of viral particles, loads, and viral RNA copies range from 1.8–3.4 copies per L of air [7,8]. Exposure with SARSCov2 and influenza occur via short-range incidence, such as face-to-face, or as in exposure through conversation and through longer-range aerosols. Although short-range transmission is easily quantifiable [9], longer-range transmission routes are difficult to quantify in various exposure scenarios [10]. Airflow and room air changes per hour are another aspect to viral loads [11]. To minimize exposure, it is recommended to increase outside airflow, improve air filtration and improve air handling equipment. Devices as simple as a fan, cardboard box and a HEPA filter is enough to improve air quality significantly [12].

### 3.2. Ultraviolet Germicidal Irradiation Engineering Controls

It has been said that sunlight is the best disinfectant. This is based on the sanitizing power of light in the ultraviolet (UV) electromagnetic spectrum, which is germicidal and virucidal. Ultraviolet germicidal irradiation (UVGI), particularly UV-C in the range of 253–260 nm, may offer added disinfection benefits. In contrast, to many disinfectant products, applied UVGI is an effective eradication method for the inactivation of microorganisms, including viruses [13] on surfaces and when combining UV with applied buffered sialic acid (SA) mixtures in a fogging system, for example, could extend the UVGI reach to hidden niches and help eradicate virus load on varied surfaces [14]. In addition, this approach can be utilized to test efficacy of various preparations for antiviral activity in general. One way we can deal with shortages in supply chain is to autoclave or steam masks and respirators in a common food steamer to sanitize them few times without too much degradation to their protection performance [15]. Further, UV-C disinfection after each use is another possible mean to effectively decontaminate masks.

Under laboratory conditions, UVGI has been shown to be effective against a suite of microorganisms including coronaviruses [16], vaccinia [17], myco-bacteria [18] and influenza [19]. In many viral outbreaks, transmission occurs through contaminated surfaces. As for SarsCoV2, a recent report from the New England Journal of Medicine indicates the coronaviridiae can persist in air for up 2 h, on copper for 4 h, on cardboard for 24 h and upwards of 2 to 3 days on hard surfaces, such as plastic and metal [20]. Because of the rapid incidence of these viral infections, there is a need for concomitant evaluation of novel engineering control methods for inactivation of viruses on surfaces. For surfaces and objects to serve as fomites and sources of viral propagation, the virus must be able to survive in association with the surface until it encounters a susceptible host. Common approaches to eradicating viruses are to use disinfectant products like bleach and quaternary ammonia (QuatNH4), which can be effective on surfaces. Recently the authors published a paper showing that positively charged coatings like quaternary ammonium polymers could be grafted on PPE surfaces, which inactivated virus and bacteria [21]. While QuatNH4 and virucidal engraftment is a viable solution, safety studies may be necessary to belay fears of use among the public. Therefore, natural virucidal coatings are likely the best option for the general population and do not require government entity (FDA, USDA or EPA) approvals for their use, as they are called GRAS (generally approved as safe) for human use and applications.

Mechanisms of UVGI disinfection for microorganisms are effective due to protein lipid and nucleic acid modification, crosslinking, and damage to vulnerable residues at the level of their primary structure because the maximum absorption wavelength of DNA and RNA molecules is around 260–280 nm. Studies have evaluated UVGI effectiveness in terms of dose and on viral nucleic acid type, namely single-stranded RNA (ssRNA), double-stranded RNA (dsRNA), single-stranded DNA (ssDNA) and double-stranded DNA (dsDNA) [22]. Viruses on a surface are more susceptible to UV-C inactivation if single-stranded (ssRNA and ssDNA) than viruses with double-stranded nucleic acid because this structure forms pyrimidine dimers within DNA and interferes with viral DNA duplication and leads to destruction of nucleic acids rendering the viruses non-infectious. UVGI is the preferred method to inactivate viruses on surfaces but poses risk to human skin and ocular tissues. Along with relative humidity, virus survival was investigated for a 90% viral reduction by Tseng and colleagues [14], who found the optimal UV dose was 1.32 to 3.20 mJ/cm^2^ for ssRNA, 2.50 to 4.47 mJ/cm^2^ for ssDNA, 3.80 to 5.36 mJ/cm^2^ for dsRNA, and 7.70 to 8.13 mJ/cm^2^ for dsDNA. They further measured the dose for all four tested viruses at 99% reduction levels were 2-fold higher than those required for a 90% viral reduction and found UV doses at 85% relative humidity were more effective than at 55% relative humidity, which suggests the hotter warmer summer months and warmers climate could offer some differential season relief during some outbreaks. UVC can generate reactive oxygen species (ROS), together with reactive nitrogen species (RNS) and reactive sulfur species (RSS) are free radicals, which are known to play a role in biological systems and can be either harmful or beneficial to the living organisms. ROS and RNS at low concentrations are well-known defense against infectious agents and viruses.

## 4. Results

### Virus Structure and Isoelectric Point (IEP) Characterization

The physical and chemical properties of the virus, when paired with specific environmental conditions can facilitate virus adsorption to fomites and dictate the time viral particles stay infectious in aerosols and droplets. Thus, the surface characteristics of a virus are important. To characterize the structural properties, we explore protein folding, hydrophobicity and surface charge. One important surface characteristic of a virus particle is the virus surface charge, which is characterized by the isoelectric point. The viral IEP corresponds to the pH where the net charge on the virus particle is zero. Since the viruses in question are contained in human oral and respiratory secretions, the pH for binding aerosolized droplets must be 4.5–6, which is close to the IEP for SARS-CoV-2. The physical and chemical properties of the virus paired with environmental conditions facilitate virus adsorption [23]. The adhesion mechanism through which viruses are adsorbed is driven by electrostatic [24] and van der Waals interactions [25], which is described by the extended Derjaguin–Landau–Verwey–Overbeek theory [26]. These interactions are also controlled by environmental factors such as pH, temperature, and humidity [27]. Further, the virus capsid protein IEP can be calculated based on the surface charges at certain pH, using the dissociation constants of charged residues using the Henderson− Hasselbalch equation [28]. Typically, the IEP is calculated using the entire amino acid sequence, regardless of the folded structure. Thus, amino acid charges that are buried in the folding of the protein are accounted for in the IEP calculation, even if those charges were not surface accessible. The calculated IEP is often close to the experimental IEP for proteins, but viral surfaces are complex and macromolecular, thus more errors are often encountered by this approach.

For now, only experimental measurements are proving accurate for IEP elucidation, as our findings suggest for enveloped viruses like SARS-CoV-2. Estimates of IEP by the amino acid sequence or sequence homology algorithms alone may be unreliable. The IEP values were obtained based on the identified proteins on SARS-CoV-2. When we compared the experimental IEP of SARS-CoV-2 to variants of interest using amino acid sequences and provided a qualitative comparison of the IEP of variants. BLAST and FASTA scans are typical search tools, which are performed on a nucleotide or amino acid sequence to impart structural information or predict function of homologous proteins. The FASTA sequence was inputted into the Protparam tool from the Bioinformatics Resource Portal ExPASy to give us the putative structural information. Conventional methods for virus IEP measurements use bulk viral solutions. Zeta potential measures the electrostatic potential difference between the electric double layer surrounding the virus particle and the surrounding solution at the shear plane as we described. Another IEP measurement is isoelectric-focusing and capillary isoelectric-focusing, which both require the fluorescently tagged viruses and pure, concentrated solutions. We developed a single-particle method to measure the IEP of virus with an atomic force microscope (AFM). The technique is called chemical force microscopy (CFM), which uses a functionalized AFM tip to measure the adhesion force of the tip and the virus immobilized on a surface. When working with our non-BSL methods and testing chamber, we utilized BEI resources for SARS-CoV-2heat-inactivated and gamma-irradiated isolates. We were found to uniquely differ, as expected. The heat-inactivated virus contained many small particles and was not further tested; it was 5.2–5.3. This is on the low end of the IEPs from the calculated sources [27].

The macromolecular structure of virus is highly dependent on the type of virus, non-enveloped or enveloped. Non-enveloped viruses have a protein capsid on their surface. One of the simplest examples is porcine parvovirus (PPV). It has 60 copies of a surface protein, with 80% being VP2 [29]. However, most of the amino acids are buried in the capsids and not surface exposed, due to interactions of the capsid’s proteins with each other and with the nucleic acids in the interior of the capsid. This makes calculation of the IEP of viral capsids even more prone to errors than for singular proteins. For enveloped viruses, the spike proteins on the outside of the enveloped (lipid bilayer) are more surface exposed than non-enveloped viruses. However, it is often difficulty to elucidate the density of the spike proteins and the exposure of the lipid bilayer. In addition, the spike proteins of enveloped viral particles are often highly glycosylated, as shown in Figure 3 [30]. The glycosylation patterns may hide many of the chemical features of charge and hydrophobicity that we desire to use in adhesion or repulsion applications. Therefore, directly applying knowledge of surface charge and hydrophobicity from molecular models and viral RNA sequences may not truly reflect the charge and hydrophobicity available on the surface. We have shown this with our work on the surface charge of SARS-CoV-2 [27].

While we can determine the exact amino acids and their glycosylation that are surface exposed, IEP is still not accurate for large biomolecules. Taking advantage of viral IEP can be one approach to repelling or attracting live virus. While evidence exists that this approach could work, little is known about the most virulent viruses and approaches to modifying masks using applied non-specific electrostatic forces could have unforeseen consequences, namely, unintended exposure. Understanding virus adsorption can help to facilitate safe practices and aid in improving PPE. For example, learning how to repel viruses from surfaces or to adsorb them could be used to improve filtration devices and build better PPE. The physicochemical properties of the virus paired with environmental conditions facilitate virus adsorption [23]. Further, we have determined experimentally the exact IEP of SARS-CoV-2 [27]. The virus surface charge can be used to evaluate the adhesion forces between the virus and a target substrate, and we can use this to predict the likelihood of virus attachment to a charged surface [31]. Moreover, we considered ways to apply electric fields to repel or attract and bind virions.

There are often large, charged patches distributed on the viral surface at the IEP. Once an accurate IEP is determined, it is likely that the distribution of charge on the surface of the virus may play a critical role in electrostatic adhesion. Work studying the adhesion of proteins to ionic chromatography resins has shown that proteins can have different charge, yet the same adhesion strength [32]. So, distribution of the charge across the protein can change electrostatic adhesion. Adding the docking of spike host cell receptor adds further dimension to the interaction characteristics [33]. Typically, molecular docking is used to test the binding affinity of the SARS-CoV-2 spike in four regions against the elucidated structure of the GRP78 substrate binding domain called β [34]. This same interaction is also seen for hydrophobic interactions in terms of added complexity. Immobilization of charged amino acids (lysine or arginine) in the middle of a hydrophobic patch can either increase or decrease the hydrophobic interaction of the entire patch [35]. It can be extrapolated that these patch effects would be increased on the surface of large macromolecules as compared to proteins. The large surfaces have many charges and hydrophobic patches that are often repeated, thus exacerbating the cumulative effect of charge, hydrophobicity, and patchiness has on adhesion. Work continues to better understand how to predict virus adhesion to electrostatic and hydrophobic surfaces through experimental and computational studies. Patchiness will continue to play a critical role in understanding how adhesion is different for different viruses. So, utilizing all of these structural aspects will help develop further any goal of prevention, inhibition, remediation or eradication of viral pathogens like SARS-CoV-2.

Many and wide-ranging viral isoelectric points have been characterized. These interaction properties are established under specific pH conditions, which can be difficult to ascertain or know fully with modeling. The surface charge can be used to design virus removal filters based on electrostatic adsorption [36,37]. One method to measure virus IEP is chemical force microscopy (CFM) [38]. We designed the CFM method to not only experimentally measure the IEP in relevant physiological conditions, but it is also single particle. This allows the variety of virus particles that are formed to be measured. It also reduces the necessity for highly purified samples, which is required for many IEP measurements. The IEP measure for SARS-CoV-2 was much lower experimentally as compared to computationally derived IEPs [27]. This demonstrates that our molecular calculations of IEP are not well suited for viral particles.

In that regard, molecular diagnostic methods were employed in unprecedented speed to first identify the novel coronavirus that was causing COVID19. By January 2020, we had the first viral sequence [39] and by March 2020, we had determined the structure of the SARS-CoV-2 spike protein and had determined that the Angiotensin1-Converting Enzyme 2 (ACE2) protein was the entry receptor into human cells [40]. This extremely fast molecular and structural understanding of this novel virus allowed the development of vaccines and therapeutics at paces never seen before. While it was important to understand these molecular interactions, the general understanding of how the virus survived on surfaces took longer to unfold and we are still developing new understanding on how SARS-CoV-2 interacts with surfaces. However, fomites are not the main route of transmission, and we turned our attention to aerosols and further unequivocal structural elucidation.

## 5. Applied IEP and Structural Characteristics to Improve PPE

One goal, which we base this report upon, was to offer PPE containing a decoy and inhibit or kill viruses by such means as cognate substrates and drugs or inhibitors. The aim was to find means to specifically bind target proteins for example to Hemagglutinin (HA) and Neuraminidases (NA) for Influenza, and the S-protein for COVID, effectively locking these viral pathogens it in a configuration through hydrogen bonding or covalent interaction that prevents fusion to cellular membranes through the receptor binding domain (RBD) of target viruses. In effect, this is a trap and a cellular decoy that would improve PPE, which is currently inadequate to protect the wearer fully. After exploring host binding by viral particles, which all work on step one through similar mechanisms by receptor binding to the cell surface on host cells and many chemical compounds, it was apparent that targeting the chemical components the virus utilizes for receptor binding and transmission would prevent infection of cells and tissues and constitute an effective prophylactic approach to emerging pandemics world-wide. In this way, we defeat and inactivate the viruses before any opportunity to infect their hosts and holding the virions in a stable configuration would help prevent bystander or self-inoculation. This work targets among such targets as ACE2 receptor and hemagglutinin through sialic acids, which are components of carbohydrate chains. In particularly, n-acetyl neuraminic acid is a glycoconjugate involved in cell–cell and cell-pathogen interactions and the main target that would be used to compete for host-pathogen interaction. The expressions of SAs are highly conserved from echinoderms to humans, where they constitute components of cell surface glycoproteins and gangliosides. They largely occupy terminal positions as individual monosaccharides or as oligosaccharides or polymers but infrequently. They are found in secreted glycoconjugates and in oligosaccharides, in blood, serum, milk or in mucus or other secretions [41].

In addition to using charge attraction and repulsion, we also considered more specific biochemical compounds, such as N-acetyl Neuraminic acid (Neu5Ac), SAs and other cognate biomimetic cellular components that viruses use to invade host cells and tissues. Hemagglutinin and Neuraminidases are involved in H1N1 influenza, and others, along with COVID-19 and SARS which use surface S-Proteins or spikes. All have a critical role in viral infection. Cell surface glycoproteins that contain a terminal SA or Neu5Ac are some of the targets of several viruses of concern. These are innocuous host cellular carbohydrate adducts on proteins that viruses exploit to enter cells and tissues. As positive controls, we will use actual lysed cellular membranes and specific drugs targeting the viral neuraminidases and ACE2. These same components will serve as human tissue and cellular mimetics when used on masks and filters. Here, they will attract, capture and otherwise inactivate viral particles, such as coronaviruses, SARS or any novel pathogen we choose. In this way, infectivity and virulence could be attenuated for deadly pathogenic species without the host being infected. For a general cytotoxic filtration fiber coating, sulfonic acid polymers and other natural coatings will be utilized, which will trap and likely kill bacteria and viruses that come in-contact with the fibers.

## 6. Electrical Engineering Approaches

The application of virucidal substrates together with repulsive electric fields is what we refer to as a protective shield approach to airborne viral pathogens (see Figure 4); they are a current ongoing focus [42]. In this section we describe our approach to improving PPE. In that regard, we propose such nanoparticles as plasmonic nanodots and nanopores, which are proven to be effective in combating COVID-19 virus and possibly other similar pathogens that could be encounterd. Nanomaterials based in a 3 or more-layer masks can be designed to generate charges and electrical signals towards viral antigens on such PPE as masks. In the simplest basic three-layer mask approach, the middle and the outer layers, where the hydrophobic conducting and semiconducting nanomaterials are used, effectively lead to accumulation of static charges. The rate of accumulation of charges increases with the increase of friction but come to a saturation level following the model of growth of charges in a capacitor per square unit of area. The capacitance of two parallel plates capacitor for example is typically proportional to the surface area and inversely proportional to the distance that separates the two plates. However, the capacitance value of the proposed quasi-capacitor can be increased by inserting nanomaterial layer(s) as dielectric medium between the two conductive plates which has a dielectric constant greater than air. The choice of this dielectric material depends on the mask material and the type of virus strain being targeted. Several matrices of the nanomaterial configuration had been explored via simulations [43].

Metal oxides like titanium, zinc and iron oxides and copper nanoparticles are often used in nanoparticle work [44]. The antimicrobial and antiviral nanoparticles are likely to be as effective as silver or gold or their alloys. Silver and gold have long been known for their antiviral and antimicrobial properties and had been used in medical devices to combat antigens. Surface plasmons based on silver and gold nanoparticles resulted in increased efficacy in combatting and neutralizing many types of microorganisms [45]. This study focused on the design of silver/gold impregnated COVID mask layers to be used as a face shield to prevent COVID infection and reinfections. In the case where plasmonic effects are not strong enough to neutralize antigens by silver or gold plasmonic layered face masks, wearable electronic devices such as mobile phones or compact portable battery sources are used to induce higher electrostatic effects with high efficiencies. While silver/gold nanostructure can be used as a disinfectant, even coated on fabrics, and included in sprays, they are not primary used in this way. However, face masks impregnated with antimicrobial nanoparticles are proven to provide additional protection against microorganisms and viruses. This reduction in the probability of the transmission and infection is significant; a thorough clinical investigation should follow to quantify and optimize their efficacies for these purposes [46].

Cost becomes an issue with silver and gold metals and impregnated material. A more economical approach is to use silver/copper alloy nanoparticles. The antiviral properties of a silver/copper alloy nanoparticle-coated masks or High Efficiency Particulate Air (HEPA) filter was evaluated in air filtration systems of aircrafts and commercial buildings. The ability of nanoparticles (gold, silver, and copper alloys) when deposited on HEPA filters to stop and neutralize coronaviruseswere investigated [47,48]. Existing HEPA filters are initially coated with silver and copper alloys nanoparticles using a spark discharge method. The method of spark discharge is of special interest, because it is fast, clean, flexible with respect to material, and easily scales-up. The nanoparticles sizes distributions are narrow, and the mean primary particle size can be controlled via the energy per spark. Separated, unassembled particles, 2–15 nm in size are obtained by controlling the flow rate.

Samples of a HEPA filter will be coated with metal nanoparticles using the low-cost copper alloys nanoparticles and their performance is compared to the more expensive silver and gold nanoparticles. Nanoparticles are synthesized via an atmospheric spark discharge method and deposited onto commercially available filters using forced convection flow [49]. A controlled number of nanoparticles were deposited on each filter sample; a variety of deposition patterns, sizes, and densities of nanoparticles were explored in parallel with mathematical models to examine the antiviral efficacies of the filter samples. The size distributions of the copper alloy nanoparticles on the filter samples are measured using a scanning mobility particle sizer to optimize the mathematical model. The mathematical model will establish the decay equations for the decline rates of the antiviral ability against COVID-19 across the modified HEPA filter samples. The model will also address the influence of several parameters on the primary particle size and mass production rates [50].

When considering attraction versus repulsion there could be bystander consequences or unintended static forces, e.g., contamination as well as intended forces, which we call pinning. Because the electric charge accumulation of the RNA viruses is much less than the electrostatic charges accumulated in the layers of the mask within a few minutes, this would minimize the bystander effect. However, if one adds a virucidal substrate or coating the bystander effect is minimized or perhaps eliminated. If dust floats near an object at high voltage, or becomes highly charged, the dust will usually be attracted and then will often stick to the object. The behavior may seem counterintuitive as opposite charges attract, same charges repel, an uncharged dust particle should be unaffected by a charged object [51]. However, viruses have charged patches and can accompany dust and change the charge distribution. Furthermore, even if dust meets the object, we might expect the dust to acquire some of the charges and therefore be repelled and not attracted. If in fact, surfaces that have very high charge are created to consist of two layers of water resistant or hydrophobic microfiber polyester or melt-blown fiber and an inner cotton layer, for example, that would repel airborne droplets, but calls into question what happens at the outer surface charged interface. The inside surfaces of the layers are brushed and meshed to create an additional barrier. We are likely inducing a dipole when dust is near a negatively charged surface, the electrons are likely to be pushed away to the far surface of the dust particle. This then makes the near surface positively charged and attractive. Biologics do not induce dipoles. However, even at its IEP, viruses still have a lot of charge on them [52,53]. They just have an even number of positive and negative charges. Therefore, they may still stick to a positively charged surface.

Contact electrification could be considered using a set of layers to create and hold charges from electrostatic interactions, also known as triboelectric charging. Here, a type of electrical charging is generated with materials, such as cotton to polyester or nylon. These layer examples become charged, usually with separation or movement of material with which they are in contact. In this regard, layers can create and hold charges. The layers could consist of the inner surface of cotton or type of non-conducting material, which is basically a moisture wick and a nonconductor so that the electrostatic charges produced inside the layers do not drain out through the surface of the face. Most electrostatic applied charges are positive, and it is suggested that virus is bound by this. Since the charges could attract virus particles, a bystander effect could be contraindicated as a main approach [54].

Since the COVID-19 comprises an amount of micro-sized droplet nuclei carrying a net electric charge, a supplemental conductive mesh layer is used to a contact DC current for a wearable voltage generator to supply negative charge to the conductive mesh [55]. The conductive mesh is made up of parallel wires properly spaced to effectively conduct the current unidirectionally. Wearable electronic devices, i.e., mobile phones, compact power bank sources will interface the face mask via a standard mini universal serial bus (USB) to provide constant power for a tunable electric field with various strength at given input applied voltages. In this parallel configuration, the electric field is uniform in the region between the wires along the mask layer, its strength varies as a function of radial position along the mask. The field generator will supply negative charges to the conductive mesh layer, which contains positively polarized micro-sized droplet nuclei; an ionic stream and negative ions are generated. In general, particles will be ionized by the negative ions and attracted to the opposite pole of the mask layer [56]. This will act as double electric wall protection against the virus making this mask layering design able to capture particles of the size of respiratory droplets and aerosols regardless of their phage concentration.

COMSOL simulation is used to estimate the electric field distribution by varying different parameters: electric field strengths, electric field directions, and in extreme cases safe electric fields and breakdown field strengths. The electrons/ions continuity equations in two- and three-dimensional models are setup using a point to plane configuration. At the interfaces, the divergence condition implies a condition on the normal component of the field and the curl condition implies a condition on the tangential component of the field [57]. The mask layers interfaces are considered discontinuities. Gauss law for the electric field is used to derive the closed surface electric flux and the Faradays law is used to derive closed contour electromotive force also known as the electromagnetic induction. The calculations of electromagnetic induction forces were useful when computing electrostatic forces and capacitance values.

## 7. Virucidal Substrate Coatings for Prophylaxis

Viruses like SARS and H1N1 can survive on fomites from 4 h up to 24 h [20] and can be carried on small airborne charged particles, such as dust. Given that the typical RNA coronavirus has a high (μ) mutation rate, evolving at an estimated rate of 10^4^ nucleotide substitutions per year [58] suggests that finding a suitable conventional vaccine over the long term may be difficult. Unless the capsid remains stable in structure as a vaccine target, other measures such as the one in this work would constitute a rapid and immediate response, while vaccines are in trial or developed further. Moreover, since these applications involve no human ingestion, they are considered generally safe for application on surfaces and PPE. Viruses typically have no net charge or are positively charged, under physiologic pH or in oral secretions, pH 5–9. Taking advantage of the many isoelectric points IEPs of viruses, under specific pH can be one approach to trapping, repelling, or attracting live virus. One does not have to tear the virus apart, just immobilize it, and otherwise inactivate, neutralize, or eradicate it. While evidence exists that this approach can work, any mask can have unforeseen consequences and unintended exposure when used with masks and filters. To avoid the risk of any bystander effect, we propose novel and molecular approaches such as virus trapping and eradicating coatings [59].

Masks and respirators are arguably the most important piece of PPE. In this section, we outline the general mask design construct. The three-layer mask design (see Figure 5) can be modified or produced to prevent the immediate infection from DNA or RNA viruses, including SARS-CoV-2 and influenza. Masks work as a physical barrier to respiratory droplets that enter through the nose and mouth from infected individuals. Virus size, infectivity level, distance from infection source and chemical properties vary and influence how well masks work. Their role is the first line of defense and are key to preventing respiratory and COVID-19 infection from presymptomatic, paucisymptomatic or asymptomatic carriers shedding virus [58]. However, mask design and tightness of fit and construction material also affects the performance of facial masks considerably. Importantly, relative susceptibility for viruses according to the CDC is increasing as follows: small non-enveloped viruses > large non-enveloped viruses > enveloped RNA viruses (such as SARS-CoV-2). Viral Filtration tests show the DermaSaver, Sci-Tex Mask material construction to be an average 91.8% effective in preventing aerosolized virus particles from penetrating through the mask [59]. Before coating and modifying current mask material a testing chamber was devised to mimic exposure (see Figure 2). The surrogate virus particles that we use to test in this construct are five times smaller than COVID-19 virus particles and less susceptible to sanitation means, which when eradicated implies higher efficacy against preventing COVID-19 and influenza virus particles.

Coatings are diverse but center around a few key concepts, namely binding or trapping agents to fix and secure virions and virucidal substrate to kill any incident particle or pathogen. For example, HA or the S-protein interact and locks in a configuration on a non-cellular surface that would prevent fusion of the cellular and viral membranes in vitro. In effect this is a trap and a cellular decoy. Below is an example list of targets to achieve this goal by type and include N-Acetylneuraminic acid (Neu5Ac), NA inhibitors like Oseltamivir, O-GlucNAcase inhibitors like Thiamet G and sugar analogues like disaccharide or polysaccharide decoys bound on a matrix scaffold to mask material. Utilizing viral binding moieties to trap coronavirus, we turned to sugars, glycans carbohydrate polymers and matrigels, which are the first innocuous and effective coating to consider, and evidence suggests *N*-acetylated (GlcNAc)/*N*-sulfated glucosamine (GlcNS) and glucuronic acid (GlcA) are a able to capture viral spike proteins [60]. Further, possible inexpensive version of our mask concept may lie with sugar linked polymers of seaweed, which is an inexpensive and abundant renewable source of these glycans. This renewable abundant natural plant-based approach holds promise for many or humanity’s needs. This is a negatively charged matrix, which was shown to worked on viruses. Since NA/Neu5Ac is the predominant sialic acid found on human and many mammalian cells, choosing these as target decoys for PPE coatings make tremendous sense from safety and efficacy standpoints. Other forms of sialic acid-type decoys include N-Glycolylneuraminic acid, which also occur on cells and could be useful surrogate targets for cornoviridiae or for binding approaches across the viral pathogen spectrum. These residues are negatively charged at physiological pH and are found in complex glycans and in chains on mucins and glycoproteins at the cell membranes and give these fluids a slippery feel. In fact, N-Glycolylneuraminic acid, a particular sialic acid, is known to act as a decoy for invading pathogens. Along with involvement in preventing infections, Neu5Ac acts as a receptor for influenza and other viruses, allowing attachment to mucous cells via hemagglutinin an early step in acquiring viral infection. Moreover, sugars and glycans from renewable and sustainable sources like seaweed are extremely viable options for new mask constructs.

Seaweed utilization as raw material for the production of hydro- colloids—agar, carrageenan from red seaweed and alginates from brown seaweed, which are used in food and cosmetics industry and also in medicine and pharmaceuticals. It is the features of biologically important N-glycans, N-glycopeptides N-glycans linked to glycoproteins found in various seaweed species that have been identified, characterized and of which we take note [61]. Diverse forms of algal seaweed is largely classified into 3 groups based on color pigmentation. Genera of agar-containing red seaweed *Gelidium* sp. and *Gracilaria*; genera of carrageenan-containing red seaweed *Chondrus crispus*, *Gigartina*, *Iridaea*, *Kappaphycus alvarezii* and *Euchema denticulatum* are used most widely [62]. Only high-mannose type N-glycans occur in seaweed glycoproteins present in either algae or seagrass. Griffithsin a 12.7 kDa lectin found in the Griffithsia genus (red algae), is one of the most promising inhibitors of MERS-CoV [63]. Except from the best-known species, mentioned above, *Cladophora* spp. of green algae is used for purpose of hydrocolloid production and some species of seaweed may contain native toxic compounds [64].

Support for NA and GA use comes from development of inhibitors of enzymatic cleavage of host sugars, which have been in trial and are effective at preventing infections for some viral species, but not all. Interestingly, these approaches also include targets of sialic acid residues. Neuraminic acid inhibitors are based on enzymatic hydrolysis of the sialic acid (Neu5Ac)-terminated glycoprotein. Two possible strategies have been applied through devising congeners or the related compounds having comparable chemical structures and biological functions, and the conjugate, which are compounds involving two bioactive substrates joined through covalent bonding [65] Small molecule inhibitors that block any of these steps can produce control strategies and prevent influenza or coronavirus infection [66]. For example, the influenza hemagglutinin trimer mediates the attachment to host cell glycoproteins containing a terminal sialic acid residue through glycosidic bonds to galactose or other sugars. Avian Influenza viruses recognize the 2,3-linked Neu5Ac host receptor, whereas the human-derived viruses recognize 2,6-linked Neu5Ac receptor [67]. Zanamivir, which is structurally like the natural substrate Neu5Ac and rarely induces resistant viruses is one example. Neuraminidases, especially, focuses on using congeners. We suggest a call into production significant quantities of these sialic acid compounds for abundant use. Further, we apply them as coating agents for objects, such as masks and gowns or surfaces as an effective preventive measure for patients and pets alike. In that regard, a one pot synthesis has been devised for abundant production. Neu5Ac can be prepared from natural sources by hydrolysis [64], enzymatic conversion [68,69,70] or chemical synthesis. Among these methods, enzymatic conversion catalyzed by N-acetyl-D-glucosamine (GlcNAc) 2-epimerase (EC 5.1.3.8) and Neu5Ac aldolase (EC 4.1.3.3) is preferred [71].

Natural compounds to inhibit, inactivate and kill COVID-19 are likely future considerations, as is evaluating tissue mimetics and agents known to block, trap, inhibit or inactivate aerosolized viruses, before they infect their host, would be a technological leap forward for PPE and disease management. To delineate mechanisms, predict interruptions to common infection features and suggest improved personalized treatment is the expected outcome. Moreover, adding virucidal coating could further enhance the trapping and offer safe means of eradication with such substrates as sulfones and thiocyanate or peroxide-based disinfectants or virucides. At physiological pH, hypothiocyanite (produced in this system) is believed to mediate the oxidation of essential bacterial amino acids and the SH group of the enzyme to suppress bacterial activity. There is also a theory that lactoperoxidase is involved in the formation of higher oxyacids of thiocyanate ions such as cyanosulfite and cyanosulfate, which are also thought to have antibacterial effects that suppress bacterial activity [72]. The importance of electrostatic viral interactions can be illustrated in the early phases of vesicular stomatitis virus infection [73].

Unfortunately, the antibacterial or antiviral mechanism of the thiocyanate/peroxidase/hydrogen peroxide system has not yet been accurately confirmed. We proposed the use of toluene sulfonic acid polymers such as those used currently in preparation of ion exchange residues and unprotonated p-toluene-sulfonic (TpOH) acid matrigels as another less safe but highly effective approach to the hydrophobic outermost layers of a multi-part mask. The key to this approach is to use the column chromatography chemistry to fix, bind or otherwise adhere the sorbent substrate to the outer parts of the PPE to immobilize it and make it permanently stable. The toluene substituents withdrawing electron density from the aromatic ring. An example of this is the nitration of toluene during the production of trinitrotoluene. Toluene is used extensively as a solvent in the lacquer industry, and in the explosives industry it plays an important role as a starting material for trinitrotoluene [74]. Important to note that Toluene sulfonic acid is used as a skin exfoliant and when dry, grafted or unable to be aspirated, they should pose minimal risk to nasal passages or mucous membranes or risk of ingestion.

The electrostatic binding potential of several promising natural substrates, such as small phenolic molecules (≤2000 Da), Safer coatings are key to public acceptance and many of the natural products and COVID-19 drugs that inhibit viral docking and entry are now known to be almost exclusively small phenolic and polyphenolic compounds. Applying these compounds to our novel assay system (see testing chamber), we expect small phenolic molecules will act as a decoy or otherwise bind and trap many viruses. Most of these small phenolic molecules are bioactive. This provides a potential mechanism through which chemicals, such as sialic acid or sulfonic acid, fixed to select proteoglycans, Matrigel and alginates could also inactivate or kill viruses. We propose to survey, test and evaluate these small phenolic molecules in relation to their trapping or inhibiting properties. Small phenolic molecules have been previously associated with antiviral and inhibiting activity. However, attempts to define their clinical implications were hampered by experimental constraints and sometimes toxicity. The available data on use of phenolic acids with COVID19 is largely unexplored and mechanistic explanations are often lacking. Advances in structural elucidation the RBD of many viruses and a reconceptualization of PPE demand a reevaluation of these small phenolic molecules and matrices.

There are in addition, multiple other potential matrices, mostly natural products, which studies have shown mediate inhibition of viruses. The fact that Glycosaminoglycans (GAGs) in particular. heparan sulfate (HS), and heparin (Hp), which are the most highly charged macromolecules in animals suggests that electrostatic forces play significant roles in modulating their protein affinity. Such long-range interactions are typically relegated to supportive roles in protein recognition and one reason specificity and hydrogen binding becomes key in trapping and immobilizing the viruses in place. Key charged proteoglycans, seaweed and heparan sulfate are associated with inhibition of COVID-19 and other virus. In preliminary exploration, several types of proteoglycans and sulfated matrices have been shown to inhibit COVID-19 in vitro. However, multiple possible confounds remain to be adequately resolved. Accordingly, we also will conduct a combinatorial exploration of a wide range of small molecules in hydrogel coated melt blow fibers.

## 8. Conclusions and Future Considerations

The presenting problem in early 2020, now known as the days of COVID-19, was how does one address emerging viral threats ahead of any available vaccines, without engaging rolling draconian lockdowns. Furthermore, complications arose when science was targeted by disinformation campaigns. This resulted in a significant change in attitude towards wearing masking and taking vaccines. Virology and public health requests and mandates were largely ignored, and science was found out of favor by some citizens s refuse to respect science or comply with physical restrictions. Efforts to change behavior about PPE failed and in these select countries, the behavior put at risk individuals with advanced age or compromised health. Moreover, when these approaches fail, as they did, perhaps having a superior product with improved function could have changed the outcomes and saved countless lives. One then, logically, turns to environmental and engineering approaches for pandemic remediation, especially when no vaccine is available. Employing highly effective virus-trapping and eradicating measures, including other engineering approaches, such as better air handling, ventilation and filtration that addressed early viral pathogen community spread could have helped ended the pandemic and this calls for future preparedness, now that all is believed righted.

The development of improved PPE has been identified as a critical unmet need. Moreover, environmental science, which is more reliable than improving PPE and general virucidal approaches seem to be the more affective intervention human expectation of behavior change of compliance and follow-through suggests our approach would be a superior and more efficacious endeavor, coupled with educational considerations. Our progress toward improving PPE was severely hampered by disinformation and an interrupted supply chain. However, to date, these issues seem to have been only partially resolved. We suggest our approach will guide biomaterial development and greatly improved or enhanced PPE. Nevertheless, vaccine hesitancy and limited available treatment options remain an issue to improving outcomes. Therefore, basic understanding of SARS-CoV-2 infection and remediation is advancing. However, what is required is a clear stance of future viral pandemic preparedness, regardless of mechanisms of viral transmission. Beyond the immediate necessity for PPE and EPPE, the approach outlined in our project would aim to benefit the world against emerging pandemics and current or future viral outbreaks or resistant variants. For example, these approaches would work against the new strain of H1N1 swine flu (G4 EA H1N1), previously discussed. Getting up to speed to confront the crisis that now or will present itself and will involve innovative approaches that can be immediately applied. Our COVID proposals, which this contribution encompasses. Further, the approaches discussed on natural product coatings would not require immediate safety testing, as no ingestion would occur with a person or pet and would be safer if it happened as many of these compounds are currently in use or tested and approved as drugs and viral inhibitors. The off-use application could certainly be waived for emergency use authorization. The goal of these application has been to offer a biologic decoy to inhibit and or kill viruses by such means as cognate substrates or drugs and inhibitors can be made to bind virions tightly to the substrate and the coating are also fixed to the mask layers. For example, HA or the S-protein can lock virons in a configuration that prevents fusion of the cellular and viral membranes. In effect Authors contribute this is a trap and a cellular decoy. We hope this work is read and shared for the sake of future mankind over funding.

## Figures and Tables

**Figure 1 microorganisms-10-02407-f001:**
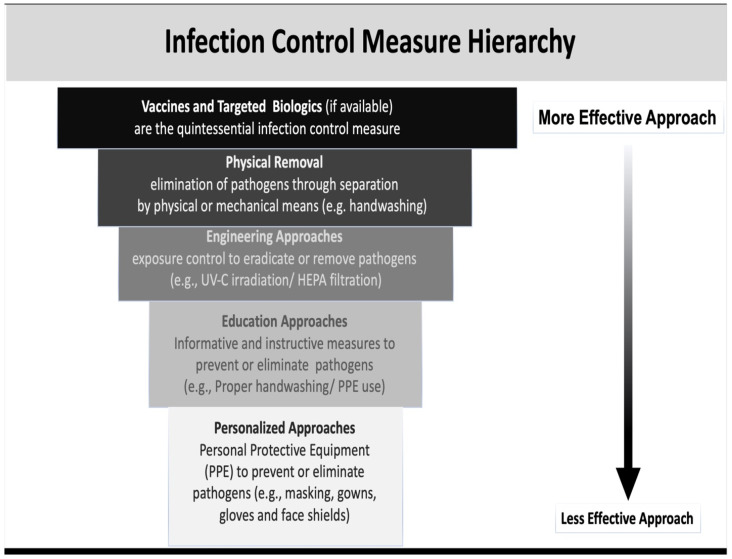
Infection control hierarchy. Adapted from Centers for Disease Control topics of control 2015. Showing approaches established to be effective in eliminating viral pathogens.

**Figure 2 microorganisms-10-02407-f002:**
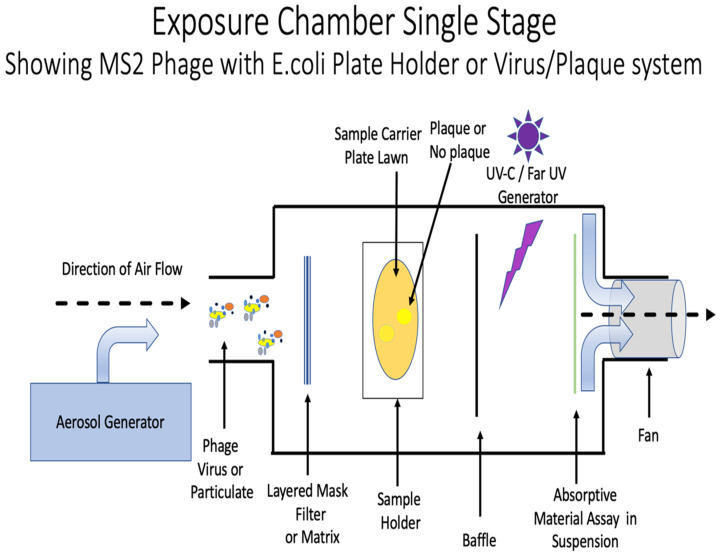
Testing and exposure chamber designed with MS2 phage and other viroids.

**Figure 3 microorganisms-10-02407-f003:**
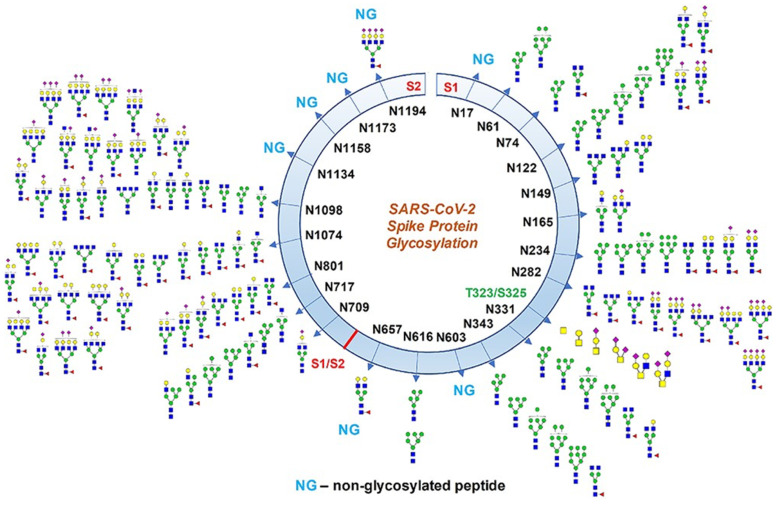
Glycosylation profile on coronavirus SARS-CoV-2 characterized by high-resolution LC-MS/MS. About 17 N-glycosylation sites were found occupied out of 22 potential sites along with two O-glycosylation sites bearing core-1 type O-glycans. Some N-glycosylation sites were partially glycosylated. Monosaccharide symbols follow the Symbol Nomenclature for Glycans (SNFG) system). This figure was reprinted and adapted permission from with the 2015 paper by Vari et al. [30].

**Figure 4 microorganisms-10-02407-f004:**
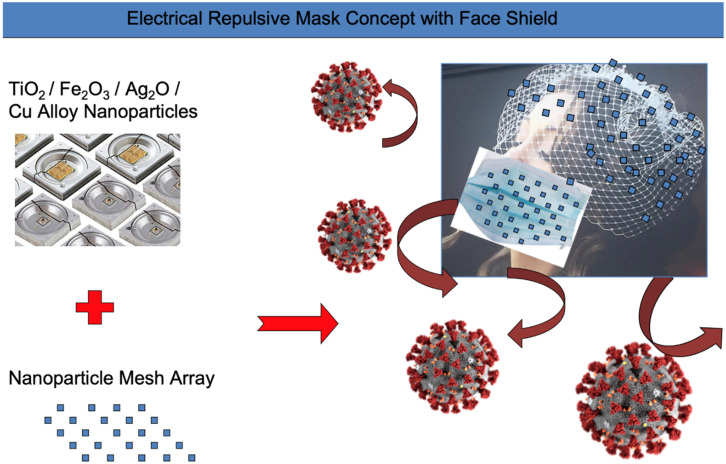
Electronic shield and repulsive field mask design with nanoparticles.

**Figure 5 microorganisms-10-02407-f005:**
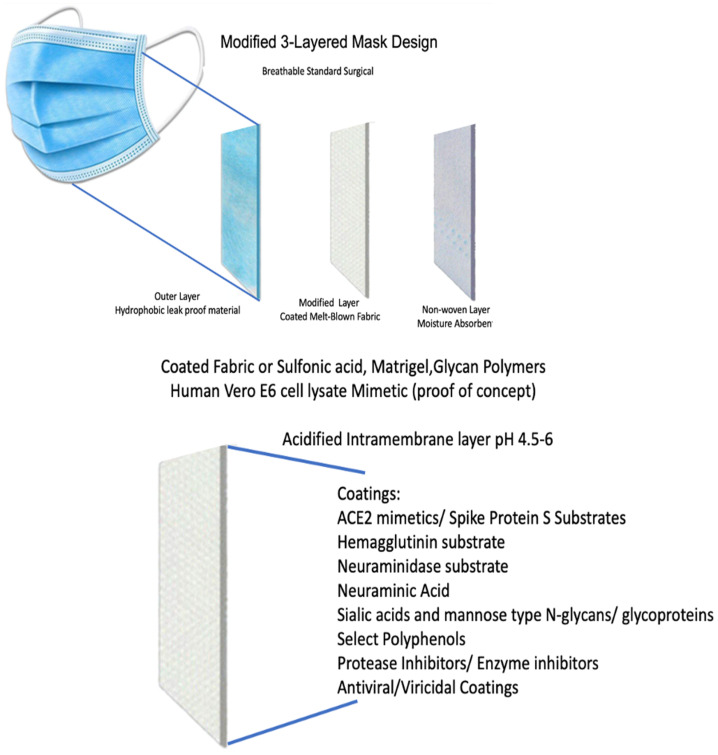
Three-layer mask with melt-blown-fiber, modified coating layer and vapor barrier.

## Data Availability

Data is available to interested parties upon request and legsl review or approval.

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
