# Peer review of "Prophylaxis and Remediation for Future Pandemic Pathogens—(Lessons from a Post-COVID World)"

_microorganisms, 2022, doi:10.3390/microorganisms10122407_

Round 1
Reviewer 1 Report
This review needs major reorganization of the piece. While subtitles are employed they do not guide the reader well. The thrust of the paper is not at all clear until the 3rd para of the introduction. The preceeding rationales are of interest but could be summed in the conclusion.
On page 8 you present a schema for interrupting viral transmission which could be used to frame the paper. You are not clear on fomites versus respiratory interruption until mid way through the paper.
The extensive discussions of isoelectric points deployment as a strategy, and decoy deployment need to be firmly nested in the overall structure of your arguement for more and better PPE. They are complex arguements and your paragraphs are long and complex. Each paragraph should begin with a statement (which you often bury in the middle of scientific facts in your paragraphs) so the reader can follow. The paragraphs are too long for this reader...chop them up and make them more concise.
So not mix supply chain issues into the basic science. Would advise a separate section at the end to group these.
Environmental science has always shown that changing structural parameters is more reliable than relying solely on behavioral change for prevention and you present some interesting options. However the text does not clearly allow the reader to appreciate these.
Your conclusion is weak. It needs to be reinforced by taking the key messages from each of the previous paragraphs and developing the overall framework. This could be a better place for the first two paragraphs of the introduction to wrap up.
Author Response
We wish to thank the reviewers for their assistance with our manuscript.
We have addressed the issue of fomites vs other engineering approaches, which was one part of an overall strategy that we address as part of a complete introduction see yellow highlights and newer sections added.
To be clear, elimination of fomites, which involves basic physically removing pathogens, is accomplished through surface disinfectant application or handwashing coupled with engineering controls, like UV-C germicidal light and air filtration [3]. The focus of this manuscript is to utilize select environmental engineering approaches to target airborne transmission, over fomite transmission, which is one part of an overall strategy to limit infection risk indoors and on surfaces. When combined with instruction for correct PPE removal or handwashing technique, which are termed administrative or educational controls and are increasingly less effective as we work closer to the individual transmission level and personal infection control.
To address the issue of adequately describing method, we have moved our methods to a section termed “methods and materials”
Further, we have added background and significance as well as a results and conclusions section and references.
To address the issue of clear conclusions and results we have added a section results and conclusions and future directions at the end of the manuscript, moving the introduction parts to the end as a summary instead of an introduction or background setting as requested.
To address the issues of sentence run on in electronic and the IEP sections. We have made some changes and thoroughly reviewed the text so as to ease the reader’s effort and understanding (see yellow highlights) for changes. We have added references in yellow where requested.
To unconfuse we have removed the issues of supply chain on the science and background and offer a one pot synthesis as a scientific approach to the supply chain problem.
Your point on scientific structural change vs relying on patient compliance and follow through is well taken and we address this in the conclusions.
We have addressed the issues of weakness of conclusion and added more relevant information to the conclusion. We have added a recap of each of the previous paragraphs to drive home the strength of the approach.
Reviewer 2 Report
The article is quite interesting. I wonder if it is a perspective paper or a review paper.
1. The author(s) should write what the aim of this article is. Methods should also be described shortly.
2. The data in Figure 1 are quite interesting. However, the question is how the effectiveness was measured. What does "effective approach" mean? It could be described in the main text.
3. There are quite long parts of the paper where none of the publications are cited (part 5. Electrical engineering approach, p. 10).
4. Conclusion is definitely needed as part 6.
Author Response
We wish to thank the reviewers for their assistance with our manuscript
The manuscript is taking a project report format and has some components of a review for the purpose of background and significance.
We have rewritten the aim of this manuscript early on and briefly described our methods.
We have added 6 new references into the body of the text that were missing.
Finally we added a conclusions and future direction section
Round 2
Reviewer 2 Report
Th authors have revised the manuscript and from my point of view it can be published now.